# FedMobile: Enabling Knowledge Contribution-aware Multi-modal Federated Learning with Incomplete Modalities

## Abstract

The Web of Things (WoT) facilitates interoperability across web-based mobile and ubiquitous computing platforms and application domains, aiming to complement and preserve existing IoT standards and solutions. In this context, the multimodal federated learning (FL) paradigm has been introduced to enhance WoT by enabling the fusion of multi-source mobile sensing data while preserving privacy. However, a critical challenge in web-based mobile sensing systems employing multimodal FL is modality incompleteness, where certain modalities may be unavailable or partially captured, which can adversely impact the performance and reliability of these systems. Current multimodal FL frameworks typically train multiple unimodal FL subsystems or apply interpolation techniques on the node side to approximate missing modalities. However, these approaches overlook the shared latent feature space among incomplete modalities across different nodes and fail to discriminate against low-quality nodes. To address this gap, we present FedMobile, a new knowledge contribution-aware multimodal FL framework designed for robust learning despite missing modalities. FedMobile prioritizes local-to-global knowledge transfer, leveraging cross-node multimodal feature information to reconstruct missing features. It also enhances system performance and resilience to modality heterogeneity through rigorous node contribution assessments and knowledge contribution-aware aggregation rules. Empirical evaluations on five widely recognized multimodal benchmark datasets demonstrate that FedMobile maintains robust learning even when up to 90% of modality information is missing or when data from two modalities is randomly missing, outperforming state-of-the-art baselines. Our datasets and code are available at the link.

## CCS Concepts

• **Computing methodologies** → *Distributed algorithms*; **Distributed artificial intelligence**.

## Keywords

Multi-modal Federated Learning, Incomplete Modalities, Web-based Mobile, Knowledge Distillation, Model Aggregation

**ACM Reference Format:**
Anonymous Author(s). 2018. FedMobile: Enabling Knowledge Contribution-aware Multi-modal Federated Learning with Incomplete Modalities. In . ACM, New York, NY, USA, 12 pages. https://doi.org/XXXXXXX.XXXXXXX

## 1 Introduction

In the Web of Things (WoT) [8, 28, 43], multimodal mobile sensing systems enhance the interoperability and usability of web-based mobile platforms by integrating data from multiple sources [6, 36, 42]. In this context, these systems boast a diverse array of real-world applications [25], frequently deployed to address complex tasks within domains such as autonomous driving [48], mobile healthcare [9], and the Internet of Things [18]. In this context, these tasks often prove too intricate and dynamic to be effectively tackled solely through reliance on a single sensor modality [14, 24]. Consequently, a straightforward approach involves aggregating complementary modality data from multiple sensors to extract feature information across various sensing channels, thereby enhancing model performance [13, 29, 38]. However, this multimodal learning paradigm, reliant on centralized processing and aggregation of raw user data, introduces significant privacy concerns [5, 34, 35, 45].

To mitigate the privacy concerns outlined above, Federated Learning (FL) [21] emerges as a solution. FL, characterized as an evolving privacy-preserving distributed machine learning paradigm, facilitates collaboration among mobile sensing devices across regions without compromising privacy [12]. By sharing model updates instead of raw data, FL fosters collective learning of global models among geographically spread devices. Although most FL methods deal with unimodal data for tasks like next-word prediction [10], some applications, *e.g.*, Alzheimer's detection [26], necessitate combining data from diverse sources (multimodal data). This has led to the development of multimodal FL systems [7] tailored for efficient processing of data from various sensory channels.

While the multimodal FL system addresses some challenges in multimodal data processing, it still suffers from incomplete sensing modalities [15, 26, 40], as shown in Fig. 1. For example, in mobile healthcare, sensor modalities often become unavailable due to sensor failures or malfunctions. This increases the variability of available sensor modalities across different nodes during runtime [26]. Thus, aggregating model updates in multimodal FL systems with incomplete sensing modalities becomes very challenging due to the varied distributions of modality types across different mobile nodes. This modality heterogeneity also intensifies model disparities between nodes, affecting the accuracy and convergence of FL [15, 40]. Existing multimodal FL methods use techniques like data interpolation [48] and modal fusion [7] to address these issues, but there is still a significant gap in efficiently utilizing cross-node modal feature information and selecting high-quality data nodes.

**Our Contributions.** In this paper, we introduce FedMobile, a novel knowledge contribution-aware multimodal FL system designed specifically for mobile sensing applications with missing modalities. Unlike existing multimodal FL methodologies [7, 26, 40], which typically focus on training multiple unimodal FL subsystems concurrently, FedMobile adopts a distinct approach. It aims to reconstruct the features of missing modalities by utilizing knowledge

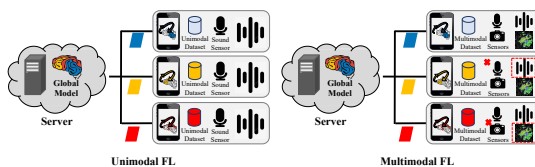

**Figure 1: Unimodal FL *vs* multimodal FL.**

distillation while introducing a knowledge contribution-aware aggregation rule via Shapley value to discern and aggregate high-quality model updates. To fulfill these objectives, FedMobile is guided by two primary goals: 1) Effectively reconstructing features of missing modalities without exacerbating modality heterogeneity or compromising main task performance. 2) Streamlining the process of identifying mobile nodes with substantial contributions while minimizing computational overhead. Next, we focus on responding to the following two challenges:

- *(C1.) How to collaboratively interpolate missing sensing modal features for different nodes with cross-modal heterogeneity.*

**(S1.)** – *The heterogeneous modality between different nodes has a common feature subspace.* In mobile sensing scenarios, malfunctioning sensor modalities at various nodes give rise to missing modalities, causing modality heterogeneity [44]. In such circumstances, conventional solutions like zero-filling [11] and parallel training of unimodal models [26] often inadequately handle this inherent feature and modality diversity. Our goal, therefore, is to leverage knowledge distillation for constructing a shared feature subspace among node modalities to improve model performance. We implement a feature generator on both the server and node levels to tackle missing modality issues. This generator, trained in a coupled training manner, aims to align different modalities by capturing a common feature space.

- *(C2.) How to find relevant metrics for measuring the contribution of a specific node in a computationally cost-friendly manner.*

**(S2.)** – *Knowledge and model updates shared between nodes and server generalize the contributions of nodes.* In FedMobile, the contributory role of participating nodes is manifested across dual dimensions: knowledge shared by local generators and local model updates shared by local nodes. To incentivize local generators to yield high quality features for incomplete modalities, we devise a novel Clustered Shapley Value approach that quantifies the individual contributions of these generators. This subsequently allows for adaptive modulation of their respective weights, thus facilitating the aggregation of high-quality feature representations. Moreover, with the objective of discerning nodes that high-quality model updates, we introduce a contribution-aware aggregation mechanism designed to retain those elements that are conducive to the overall improvement of the global model. Conversely, it eliminates nodes that do not meet this criterion. By dynamically choosing nodes based on this principle, we effectively ensure the aggregation of high-quality model updates during the training.

Additionally, we evaluate FedMobile across five real-world multimodal sensing datasets: USC-HAD [47], MHAD [23], Alzheimer's Disease Monitoring (ADM) [26], C-MHAD [37], and FLASH [30], which encompass tasks related to autonomous driving, mobile healthcare, and Alzheimer's disease detection. The results indicate that FedMobile effectively leverages various sensor types (such as GPS, gyroscopes, and radar) in scenarios with incomplete sensing modalities to accurately perform assigned tasks, even amid operational dynamics like sensor failures. Furthermore, we analyze the computational and communication overhead of FedMobile across different tasks. Extensive evaluations show that FedMobile outperforms existing multimodal FL systems (*e.g.*, FedMM [7], AutoFed [48], and PmcmFL [1]), achieving higher model accuracy with reasonable additional computational and communication overheads, especially under dynamic modality missing conditions.

The contributions of our work can be summarized as follows:

*(1)* We tailor FedMobile, a multimodal federated learning framework that is robust to incomplete modal data, for web-based mobile sensing systems in WoT.

*(2)* We design a knowledge distillation-driven cross-node modality reconstruction network to efficiently reconstruct the missing modality data without introducing excessive overhead.

*(3)* We design an efficient generator contribution evaluation module based on clustered Shapley value and contribution-aware aggregation mechanism to further improve system performance.

*(4)* We implement our design and conduct extensive experiments on 5 datasets related to 3 mobile sensing downstream tasks to explore the performance, efficiency, generality, and parameter sensitivity of FedMobile. Compared to the baselines, our approach achieves state-of-the-art performance on all tasks while maintaining comparable computation and communication overhead.

## 2 Related Work

**Multimodal Learning for Mobile Sensing Systems.** Multimodal learning aims to extract complementary or independent knowledge from various modalities, enabling the representation of multimodal data [19, 46]. This empowers machine learning models to comprehend and process diverse modal information [39]. As a result, multimodal learning techniques have become prevalent in mobile sensing, facilitating the development of systems that can understand and process diverse sensor data. For instance, multimodal learning can enhance model performance in areas such as traffic trajectory prediction [31], disease diagnosis [26], human activity recognition [3], audio-visual speech recognition [22], and visual question answering [20]. However, solving the problem of missing modalities in such systems remains an open challenge.

**Unimodal and Multimodal FL systems.** To address privacy concerns in mobile sensing systems, privacy-preserving distributed learning systems, notably FL [21, 26, 48], are emerging as a solution. FL systems can be categorized into unimodal and multimodal FL based on the number of data modalities involved. Unimodal FL focuses on constructing a global model from unimodal data while preserving privacy [27]. Similarly, multimodal FL integrates data from multiple modalities to develop an effective global model [7]. Multimodal FL systems are increasingly used in mobile sensing applications, particularly in tasks such as autonomous driving [48] and Alzheimer's disease detection [26], due to their robust multimodal data processing capabilities.

**Multimodal FL Systems with Missing Modality.** Multimodal FL systems have emerged as a promising approach for training ML models across multiple modalities while preserving data privacy.

However, in real-world scenarios, certain modalities may be missing from some nodes due to hardware limitations, data availability constraints, or privacy concerns [15, 26, 44]. To address this challenge, researchers have developed multimodal FL systems using various approaches, including modality filling [40], parallel training of unimodal models [26], and cross-model [44] techniques. For example, Xiong et al. [40] introduced a modality-filling technique using reconstruction networks, while Ouyang et al. [26] proposed Harmony, a heterogeneous multimodal FL system based on disentangled model training. However, these methods often overlook the common feature space and the evaluation of node marginal contributions, leading to issues with model accuracy. This paper aims to address these challenges by developing a knowledge contribution-aware multimodal FL system for mobile sensing.

## 3 Preliminary

### 3.1 Multimodal Federated Learning

Multimodal FL is a cutting-edge approach in machine learning (ML) that addresses the challenges of training models across multiple modalities while preserving data privacy. Formally, in mobile sensing scenarios, let us denote $\mathcal{M} = \{m_0, m_1, \ldots, m_{M-1}\}$ as the set of modalities of the local multimodal dataset $D_k$, $K$ as the number of participating mobile nodes, $n_k$ as the number of samples in the node $k$, and $d_k^m$ as the dimensionality of modality $m$ in the node $k$. The objective of Multimodal FL is to optimize a global model $\mathcal{F}(\omega)$ parameterized by $\omega$ across all modalities while minimizing the following federated loss function:

$$\min_{\omega} \sum_{k=1}^{K} \sum_{m \in \mathcal{M}} \frac{n_k}{n} \mathcal{L}(\omega; \mathbf{X}_k^m, \mathbf{y}_k), \quad (1)$$

where $\mathcal{L}(\omega; \mathbf{X}_k^m, \mathbf{y}_k)$ is the loss function for modality $m$ at node $k$, $\mathbf{X}_k^m$ represents the data samples for modality $m$ at node $k$, $\mathbf{y}_k$ is the target label associated with the samples at node $k$, and $n = \sum_{k=1}^{K} n_k$ represents the total number of samples across all nodes. In multimodal FL, the global model $\mathcal{F}(\omega)$ is updated by aggregating local model updates from each node while respecting data privacy constraints. The update rule for the global model at iteration $t$ can be formalized as:

$$\omega^{t+1} = \omega^t - \eta \sum_{k=1}^{K} \frac{n_k}{n} \nabla \mathcal{L}(\omega^t; \mathbf{X}_k, \mathbf{y}_k), \quad (2)$$

where $\eta$ is the learning rate and $\nabla \mathcal{L}(\omega^t; \mathbf{X}_k, \mathbf{y}_k)$ is the gradient of the loss function with respect to the global model parameters $\omega^t$ at node $k$. Clearly, when a modal sensor on a mobile node fails or ceases to function, resulting in a missing modality, the optimization of Eqs. (1) and (2) becomes challenging. This impediment implies that multimodal FL may struggle to fulfill the designated task effectively under such circumstances.

### 3.2 Shapley Value in ML

Shapley Value [32] is a concept from cooperative game theory used to fairly distribute the value generated by a coalition of players. In the context of ML, it is often applied to understand the contribution of each feature to a model's prediction [2]. Let us denote a predictive model as $f$, and $\Phi_i(f)$ represents the Shapley value of feature $i$ in

the model $f$. The Shapley value of feature $i$ can be computed as:

$$\Phi_i(f) = \sum_{S \subseteq N \setminus \{i\}} \frac{|S|!(|N| - |S| - 1)!}{|N|!} \left[ f(x_S \cup \{i\}) - f(x_S) \right], \quad (3)$$

where $N$ is the set of all features, $x_S$ represents the instance with only features in set $S$, $f(x_S \cup \{i\})$ is the prediction of the model when feature $i$ is added to the set $S$, $f(x_S)$ is the prediction of the model when only features in set $S$ are considered, $|S|$ denotes the cardinality of set $S$, and $|N|$ is the total number of features. The above formula computes the marginal contribution of feature $i$ when added to different subsets $S$ of features, weighted by the number of permutations of features in $S$ to the total number of permutations of all features. In fact, calculating the Shapley value directly using the above formula might be computationally expensive [2, 32], especially for models with a large number of features.

## 4 Our Approach

**Overview.** In this section, we present the proposed FedMobile framework, as illustrated in Fig. 2. First, to reconstruct missing sensing modalities, we design a knowledge distillation-driven cross-node modality reconstruction network. Secondly, to select generators with high-quality contributions, we design an efficient model contribution evaluation module based on clustered Shapley values. Finally, to further mitigate cross-node modal heterogeneity, we introduce a knowledge contribution-aware aggregation rule for robust aggregation. Next, we will introduce each functional component in detail.

### 4.1 Knowledge Distillation-driven Cross-node Modalitiy Reconstruction Network

**Impute Missing Modalities.** Different from existing works such as AutoFed [48], which focus on reconstructing local missing modalities while ignoring cross-node feature information, FedMobile aims to collaboratively utilize the common feature subspace across nodes to iteratively reconstruct the feature information of missing modalities. Specifically, to gain insight into the data distribution of missing modalities across nodes and use this understanding to guide local model training with incomplete modalities, we employ conditional distributions to characterize the modal data distribution of each node. Let $Q_k : \mathcal{Y}_k \rightarrow \mathcal{X}_k$ denote the above conditional distribution, which is tailored for each node and aligns with the ground truth data distribution. This distribution encapsulates the necessary knowledge to guide multimodal FL training with incomplete modalities:

$$Q_k = \underset{Q_k : \mathcal{Y}_k \rightarrow \mathcal{X}_k}{\arg \max} \mathbb{E}_{y \sim p(y_k)} \mathbb{E}_{x \sim Q_k(\mathbf{X}_k | y_k)} \left[ \log p(y|x; \omega_k) \right], \quad (4)$$

where $p(y_k)$ and $p(y \mid x)$ denote the ground-truth prior and posterior distributions of the target labels, respectively. Given these conditions, we employ local models to infer $p(y \mid x)$. Consequently, a straightforward approach involves the direct optimization of Eq. (4) in the input space $\mathcal{X}_k$ to approximate features for missing modalities. However, when $\mathcal{X}_k$ is of high dimensionality, this approach may lead to computational overload and could potentially disclose information about user data configuration files. Therefore,

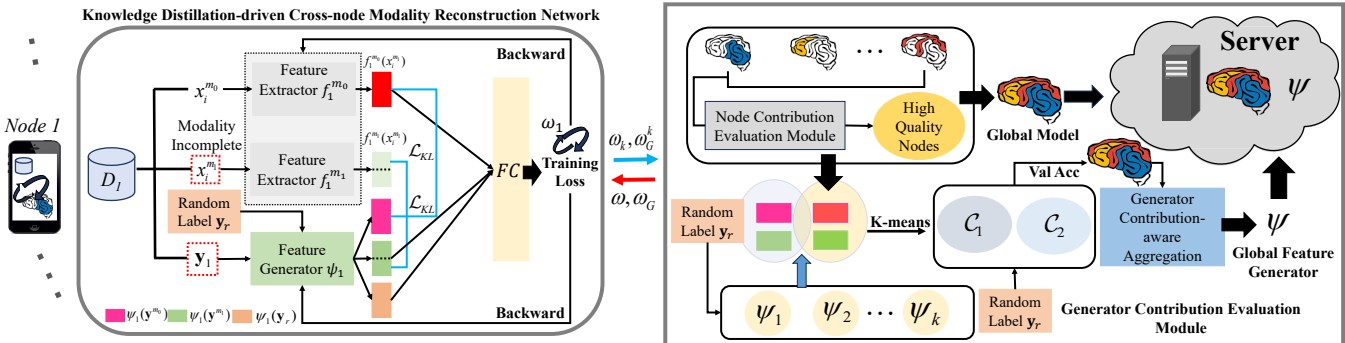

**Figure 2: Workflow overview of FedMobile.**

a more feasible alternative is to reconstruct an induced distribution $\psi_k : \mathcal{Y}_k \to \mathcal{Z}_k$ over a latent space. This latent space, being more compact than the raw data space, can help mitigate certain privacy-related concerns:

$$\psi_k = \underset{\psi_k:\mathcal{Y}_k \to \mathcal{Z}_k}{\arg\max} \; \mathbb{E}_{y \sim \hat{p}(y_k)} \mathbb{E}_{z \sim \psi_k(\mathbf{z}_k|y_k)} \left[ \log p(y|z; \omega_k) \right]. \quad (5)$$

Hence, nodes engage in knowledge extraction from missing modality data by acquiring insights from a parameterized condition generator $\psi_k$ by $\omega_G^k$. The optimization process is as follows:

$$\min_{\omega_G^k} J(\omega_G^k) = \min \mathbb{E}_{y \sim p(y_r)} \mathbb{E}_{z \sim \psi_k(z|y_r)} \left[ \mathcal{L} \left( \sigma \left( g \left( z; \omega_k \right) \right), y \right) \right], \quad (6)$$

where $y_r$ represents a set of random labels generated from the training dataset $\mathcal{D}_k$, $g(\cdot)$ denotes the logits output of a predictor, and $\sigma(\cdot)$ signifies the non-linear activation applied to these logits.

**Align Missing Modalities.** On the other hand, it is necessary to refine feature subspaces to more accurately encapsulate the local knowledge of nodes. For instance, considering a two-modality task, we can derive the generated latent space via the labels $y_k$: $\mathcal{Z}^{m_0}, \mathcal{Z}^{m_1} = \psi_k \left( y_k; \omega_G^k \right)$, where $\mathcal{Z}^{m_0}$ and $\mathcal{Z}^{m_1}$ represent the respective latent features of each modality. Assuming $m_1$ denotes the missing modality, our objective is to further empower $\psi_k$ to assimilate knowledge from various modalities, thereby enhancing the completeness and generalization of the feature space. For modality $m_0$, the learning process can be expressed as follows:

$$\mathcal{L}_{KL}^{m_0}(\omega_G^k; \omega_k) = \min_{\omega_G^k} \sum_{i=1}^{B} \mathbb{E}_{x \sim \mathcal{D}_k} \left[ D_{KL} \left[ \left( f_0 \left( x_i^{m_0}; \omega_k^{m_0} \right) \| \mathcal{Z}_i^{m_0} \right) \right] \right], \quad (7)$$

where $B$ represents the number of samples in the local training batch. For modality $m_1$, we only learn from the missing data of this modality, which is formally expressed as follows:

$$\mathcal{L}_{KL}^{m_1}(\omega_G^k; \omega_k) = \min_{\omega_G^k} \sum_{i=1}^{I} \mathbb{E}_{x \sim \mathcal{D}_k} \left[ D_{KL} \left[ \left( f_1 \left( x_i^{m_1}; \omega_k^{m_1} \right) \| \mathcal{Z}_i^{m_1} \right) \right] \right], \quad (8)$$

where $I$ represents the remaining number of samples. Finally, we use $\mathcal{Z}^{m_1}$ instead of the feature $f_1 \left( x_k^{m_1}; \omega_k^{m_1} \right)$ of modality $m_1$ for multimodal feature fusion (*e.g.*, concatenated fusion) to achieve feature alignment for missing modality. According to the above

method, the overall optimization goal of every FL node is:

$$\min_{\omega_G^k, \omega_k} \mathcal{L}_{Train}^k = J(\omega_G^k) + \mathcal{L}_{KL}^{m_0}(\omega_G^k; \omega_k) + \mathcal{L}_{KL}^{m_1}(\omega_G^k; \omega_k) + \mathcal{L}_{CE}(\omega_k), \quad (9)$$

where $\mathcal{L}_{CE}(\omega_k)$ represents the cross entropy loss of model training.

**Transfer Feature Space.** In this context, we consider the global distribution generator, denoted as $\hat{\psi}$, and the set of local distribution generators, represented by $\psi_k$ for each node $k$, as the source and target domains, respectively, in a framework of domain adaptation. This particular form of adaptation is referred to as global-to-local knowledge transfer. Conversely, the local-to-global knowledge transfer takes place at the server side. During the knowledge exchange process, the node $k$ transmits its locally generated distribution model, $\psi_k$, to the server. The server then orchestrates a guided adjustment of $\psi_k$ with the aim of systematic reduction in the discrepancy between the local and global knowledge domains through the mechanism of FL aggregation. The above process can be formalized as follows:

$$\psi = \frac{1}{K} \sum_{k=1}^{K} \psi_k. \quad (10)$$

## 4.2 Clustering Shaple Value-driven Generator Contribution Evaluation Module

**Evaluate Generator Contribution.** Considering the inherent heterogeneity of data across nodes and the varied modality missing scenarios that often arise on individual nodes, a naive aggregation of the local distribution models $\psi_k$ for knowledge transfer might inadvertently cause a general shift in the collective knowledge domain, leading to counterproductive outcomes. To mitigate this issue, we use the SV method to quantitatively evaluate the marginal contribution of each distinct $\psi_k$ to the overarching learning task. However, directly applying the SV to compute the marginal contributions of individual nodes is computationally burdensome, especially in FL scenarios involving hundreds of mobile devices. To address this challenge, we incorporate the K-means clustering algorithm to reduce the computational complexity of the SV computation. Specifically, we employ the K-means clustering algorithm to cluster the $\mathcal{Z}$ generated by $\psi_k$, resulting in multiple clusters containing $\psi_k$. We then perform average aggregation on the generator parameters in the cluster to obtain $\hat{\psi}$ as the node representative

of the cluster. In this way, we can get $\mathcal{K}$ node representatives and use these node representatives as a set $N = \{\hat{\psi}_1, \ldots, \hat{\psi}_{\mathcal{K}}\}$ in SV. Consequently, the final computation of SV can be expressed as:

$$\Phi_i(\mathcal{F}) = \sum_{S \subseteq N \setminus \{\hat{\psi}_i\}} \frac{|S|!(|N| - |S| - 1)!}{|N|!} \left[ \mathcal{P}(\mathcal{F}(\psi_S(y_r')) \cup \{\hat{\psi}_i(y_r')\})) - \mathcal{P}(\mathcal{F}(\psi_S(y_r'))) \right], \quad (11)$$

where $\mathcal{P}$ is a performance metric function, such as accuracy, F1-score, or loss, $\mathcal{F}$ represents the global model, and $y_r'$ is a set of randomly generated labels from the proxy dataset $\mathcal{D}_{proxy}$. Subsequently, the normalized $\Phi_1(\mathcal{F}), \Phi_2(\mathcal{F}), \ldots, \Phi_{\mathcal{K}}(\mathcal{F})$ is used as the aggregation weight, therefore Eq. (10) can be rewritten as:

$$\psi = \frac{1}{\mathcal{K}} \sum_{i=1}^{\mathcal{K}} \Phi_i(\mathcal{F}) \hat{\psi}_i. \tag{12}$$

### 4.3 Contribution-aware Aggregation Rule

**Node Contribution.**     To generalize node contribution in a fine-grained manner, we divide the contribution of each node in each round into local and global contributions. The local contribution represents the node's performance in that round of local training, while the global contribution represents the node's impact on model aggregation. We can calculate the local contributions as follows:

$$P_{local,k}^t = \mathcal{P}(\omega_k^t; \mathcal{D}_{proxy}), \tag{13}$$

where $\mathcal{P}$ is a performance metric function, such as accuracy, F1-score, or loss, $\mathcal{D}_{proxy}$ represents the proxy dataset, and $\omega_k$ represents the local model parameters of node $k$. It is important to note that the proxy dataset does not compromise the privacy of the training set and can be collected by the server, as is consistent with previous work [32, 41]. Furthermore, we need to traverse all nodes to calculate the above contribution. To assess how much a node's update would improve the global model, we can perform a hypothetical update by applying only node $k$'s update to the global model and measuring the global contribution:

$$\mathbf{w}_{temp,k}^{t+1} = \omega^t + \eta \Delta \omega_k^t, \tag{14}$$

$$\begin{aligned} \Delta P_{global,k}^t &= \mathcal{P}(\omega_{temp,k}^{t+1}; \mathcal{D}_{proxy}) - \mathcal{P}(\omega^t; \mathcal{D}_{proxy}) \\ &= P_{global,k}^{t+1} - P_{global}^t \end{aligned} \tag{15}$$

where $\eta$ is the learning rate. Due to computational constraints (since evaluating each node's update individually can be costly), we can approximate this by estimating the potential improvement based on surrogate metrics. Hence, we can approximate it using the node's local loss reduction metric as follows:

$$\Delta \mathcal{L}_k^t = \mathcal{L}(\omega^t; \mathcal{D}_{proxy}) - \mathcal{L}(\omega_k^t; \mathcal{D}_{proxy}). \tag{16}$$

We use $\Delta \mathcal{L}_k^t$ as a proxy for $\Delta P_{global,k}^t$. The reason we do this is that larger differences have a larger impact on the global model.

**Node Contribution-aware Aggregation.**     Upon determining the global and local contributions, we strive to incorporate them adaptively into the quality assessment process of nodes participating in model aggregation, thereby mitigating the impact of updating nodes with lower quality. To achieve this goal, we can define the aggregation weight $\alpha_k^t$ for a node $k$ as a function of $P_{local,k}^t$ and

$\Delta P_{global,k}^t$:

$$\alpha_k^t = \frac{p(P_{local,k}^t, \Delta P_{global,k}^t)}{\sum_{j=1}^K p(P_{local,j}^t, \Delta P_{global,j}^t)}, \tag{17}$$

where $p$ is a function that combines the two performance metrics. Here, an intuitive choice for $p$ is to multiply the normalized performance metrics. Thus, we normalize the Local Contribution Metrics: $\tilde{P}_{local,k}^t = \frac{P_{local,k}^t}{\sum_{j=1}^K P_{local,j}^t}$, and we normalize the Global Contribution Improvements: $\tilde{\Delta} P_{global,k}^t = \frac{\Delta P_{global,k}^t}{\sum_{j=1}^K \Delta P_{global,j}^t}$. Combining Eqs. (16) and (17), if we use local loss reduction, we have:

$$\alpha_k^t = \frac{n_k \times \tilde{P}_{local,k}^t \times \tilde{\Delta} \mathcal{L}_k^t}{\sum_{j=1}^K n_j \times \tilde{P}_{local,j}^t \times \tilde{\Delta} \mathcal{L}_j^t} \tag{18}$$

where $\sum_{k=1}^K \alpha_k^t = 1$. Therefore, we can use this weight to update the global model:

$$\omega^{t+1} = \omega^t + \eta \sum_{k=1}^K \alpha_k^t \Delta \omega_k^t. \tag{19}$$

The above process is summarized in Algo. 1 in the Appendix B.

## 5 Experiment

### 5.1 Experiment Setup

To evaluate the performance of our FedMobile system, we conduct extensive experiments on four benchmarking datasets. All experiments are developed using Python 3.9 and PyTorch 1.12 and evaluated on a server with an NVIDIA A100 GPU.

**Datasets.**     We adopt five multimodal datasets for evaluations, *i.e.*, USC-HAD [47], MHAD [23], ADM [26], C-MHAD [37], and FLASH [30] datasets. The datasets cover different modalities, objects, domains, attributes, dimensions, and number of classes, as shown in Table 7, allowing us to explore the learning effectiveness of FedMobile. To simulate an environment characterized by incomplete sensing modalities, we adopt a random selection methodology to identify a target mode from the local dataset, which will represent the state of incompleteness. We then proceed to randomly eliminate a predetermined proportion of the modal data, thereby simulating the phenomenon of missing information. Note that we distinguish between the small-scale node and large-scale node scenarios according to the scale of users (*i.e.*, nodes) involved in the dataset. In the dataset used, FLASH will be evaluated in the large-scale node scenario. More details can be found in Appendix C.1.

**Models.**     When processing ADM dataset, we harness the TDNN for audio feature extraction and combine it with CNN layers for radar and depth image feature extraction. For USC, MHAD, and FLASH datasets, a 2D-CNN model is utilized to process accelerometer data, whereas a 3D-CNN architecture is employed to analyze skeleton data. Finally, when working with the CMHAD dataset, we exploit a 2D-CNN architecture to derive video features, while 3D-CNN layers are used for extracting features from inertial sensors.

**Parameters.**     For the ADM dataset, we set the learning rate at 1e-3, with a batch size of 64. Regarding the USC dataset, the learning rate is 1e-6, and the batch size is 16. For the MHAD and FLASH

datasets, the learning rate is 1e-3, with a batch size of 16. When working with the CMHAD datasets, we maintain a learning rate of 1e-4, alongside a batch size of 16. Throughout this experiment, we utilize the SGD optimizer with a momentum of 0.9 and a weight decay of 1e-4. We set the total number of nodes $K = 10$, local epoch $E = 5$, global epoch $T = 100$, and node participation rate $q = 100\%$. We set $\mathcal{K} = 5$ in the K-means algorithm. We use a multilayer perceptron as our generator (see Appendix C.3).

**Baselines.** To make a fair comparison, we employ FedProx [16], FedBN [17], FedMM [7], PmcmFL [1], Harmony [26], and AutoFed [48] as baseline methods. Among these, the first three techniques require adaptation to cope with scenarios characterized by incomplete modalities. This adaptation is achieved through the integration of interpolation techniques, namely zero padding (ZP) and random padding (RP), which are incorporated into the FedProx, FedBN, and FedMM baselines. This augmentation enables us to gauge the effectiveness of FedMobile in dealing with heterogeneous modalities. On the other hand, PmcmFL, Harmony, and AutoFed are multimodal FL solutions that naturally cater to situations involving incomplete modalities without needing further modifications to their methodologies. Thus, we can directly assess FedMobile's learning performance in similar contexts using these baseline methods. Note that all comparison results are the average of five repeated experiments to eliminate the effect of randomness.

**Metrics.** To assess the performance of our proposed method and benchmark it against the baseline approaches, we employed accuracy as the evaluation metric, a convention that has been widely utilized in prior research [4]. To quantify the computational overhead, we tracked the aggregate time consumed in uploading and downloading models for all participating nodes throughout the FL training process, as was previously done in [30]. And we computed the cumulative GPU usage across all nodes engaged in the FL training phase. For communication overhead, we perform a fair comparison by calculating the model updates that need to be transmitted for 100 rounds of global training.

## 5.2 Numerical Results

**Research Questions.** In this section, we aim to answer the following research questions:

• (RQ1) How effectively does FedMobile, along with its respective baseline methods, fare in handling diverse and complex scenarios characterized by incomplete modal environments?

• (RQ2) How does FedMobile demonstrate computational and communicational efficiency in its running processes?

• (RQ3) How does FedMobile perform in heterogeneous data scenarios, especially in dynamic modality missing scenarios?

• (RQ4) How does FedMobile perform in scenarios with large-scale nodes and missing modalities?

• (RQ5) What are the capabilities of FedMobile in terms of multimodal feature extraction, and how proficiently can it harness and integrate features from multiple modalities?

• (RQ6) How do the individual components of the FedMobile framework contribute to its overall performance, and what specific impact do they have on its effectiveness?

**System Performance (RQ1).** To address RQ1, we perform an extensive evaluation of FedMobile, along with its comparative

baseline algorithms, using four benchmark multimodal datasets. To assess performance under diverse levels of modal data loss, we introduce a set of modality missing rates designated as $\beta = \{20\%, 40\%, 60\%, 70\%, 80\%, 90\%\}$. The experimental results demonstrate that FedMobile outperforms all other baseline algorithms consistently across all these varying degrees of missing modality data, as clearly depicted in Table 1. Notably, FedMobile showcases a 1.9% improvement relative to the current state-of-the-art baseline, AutoFed, specifically in the MHAD dataset with $\beta = 80\%$. These enhanced results stem from FedMobile's innovative strategy, which entails reconstructing modal features across nodes and tactically selecting nodes with high-quality contributions. By discovering a shared feature subspace among distinct missing modalities, FedMobile efficiently reconstructs features and simultaneously excludes nodes with inferior-quality data, thereby boosting the performance.

To further evaluate the performance of FedMobile, we tested it under a more challenging scenario involving the absence of two modalities (i.e., two-modal data missing). Specifically, we randomly omitted two modalities in a fixed ratio within the ADM dataset, which consists of three modal data, and maintained this missing configuration throughout the training process. The numerical results, recorded in Table 2, demonstrate that FedMobile continues to deliver excellent performance, outperforming other state-of-the-art baselines, including an average 4.3% improvement over AutoFed on the ADM dataset. Additionally, we observed that existing methods struggle with missing data across multiple modalities, as they heavily depend on sufficient modal information to reconstruct the missing data. FedMobile, on the other hand, does not require this, making it more robust in handling such scenarios. Additionally, we provide a privacy analysis in Appendix A.

**Computational & Communication Overhead (RQ2).** To address RQ2, we systematically document and analyze the communication cost, local running time, and GPU usage of all examined methods on the USC dataset with $\beta = 60\%$. Note that since AutoFed and Harmony also include hardware equipment, they are not included in the comparison. For the convenience of comparison, we record the communication overhead of 100 global training rounds and ignore factors such as the network environment. First, while the introduction of the generator does cause additional communication overhead, this overhead is acceptable. Specifically, the additional overhead caused by the generator is 1.65 MB for each training round. Furthermore, compared to baselines such as FedProx, which do not introduce much additional overhead, our method only adds an additional 9.02% communication overhead, as shown in Fig. 3. In performance-critical multimodal services, a small amount of additional communication overhead is acceptable because it improves the quality of service (i.e., accuracy), which is a performance-overhead trade-off. The results depicted in Figs. 5–6 and Table 3 show that the GPU utilization and local running time of FedMobile consistently remains lower than or close to that of the comparative baseline methods. This indicates that our approach does not appreciably increase local computational overhead. Given that servers typically operate as resource-rich cloud infrastructure, computations related to the server-side SV calculation do not impose any significant extra computational load.

**Data Heterogeneity Scenarios (RQ3).** To address RQ3, we eval-

**Table 1: Numerical results of system performance.**

| Datasets | $\beta$ | FedProx+ZP | FedProx+RP | FedBN+ZP | FedBN+RP | FedMM+ZP | FedMM+RP | PmcmFL | Harmony | AutoFed | FedMobile |
|---|---|---|---|---|---|---|---|---|---|---|---|
| MHAD | 20% | 75.3 | 77.0 | 75.3 | 77.9 | 78.0 | 76.7 | 78.0 | 76.5 | 77.6 | **78.4** |
| | 40% | 75.2 | 75.9 | 74.5 | 75.3 | 77.5 | 74.1 | 77.3 | 76.7 | 76.8 | **77.9** |
| | 60% | 75.3 | 75.7 | 75.3 | 75.2 | 76.5 | 74.6 | 76.5 | 75.9 | 75.8 | **76.5** |
| | 70% | 75.2 | 75.0 | 74.9 | 74.9 | 76.1 | 74.6 | 76.2 | 75.8 | 75.4 | **76.5** |
| | 80% | 74.9 | 76.1 | 74.1 | 75.4 | 76.2 | 74.3 | 76.0 | 74.3 | 76.1 | **78.0** |
| | 90% | 75.0 | 75.1 | 74.2 | 74.9 | 75.8 | 74.5 | 75.8 | 74.5 | 75.4 | **76.8** |
| USC-HAD | 20% | 56.6 | 56.6 | 58.6 | 57.3 | 58.2 | 58.4 | 59.1 | 58.7 | 60.4 | **61.1** |
| | 40% | 58.7 | 57.2 | 56.9 | 57.5 | 56.2 | 58.0 | 58.6 | 57.9 | 58.7 | **62.8** |
| | 60% | 57.3 | 58.2 | 57.0 | 57.6 | 57.2 | 58.6 | 57.9 | 58.7 | 60.1 | **61.0** |
| | 70% | 57.2 | 57.9 | 57.3 | 57.0 | 57.5 | 57.8 | 58.2 | 57.1 | 58.4 | **59.8** |
| | 80% | 57.5 | 56.3 | 57.6 | 57.1 | 56.9 | 57.5 | 58.0 | 56.8 | 58.2 | **58.8** |
| | 90% | 57.8 | 57.7 | 57.1 | 57.3 | 57.1 | 57.8 | 58.1 | 56.8 | 57.6 | **59.6** |
| ADM | 20% | 82.5 | 83.5 | 83.0 | 82.4 | 82.5 | 83.1 | 83.6 | 82.8 | 83.9 | **84.9** |
| | 40% | 82.1 | 82.6 | 82.2 | 83.0 | 82.7 | 83.3 | 84.0 | 83.4 | 83.6 | **84.3** |
| | 60% | 81.3 | 81.0 | 82.4 | 82.8 | 80.0 | 81.8 | 83.8 | 82.9 | 83.8 | **84.4** |
| | 70% | 81.4 | 82.1 | 81.7 | 81.2 | 81.5 | 82.3 | 82.8 | 81.6 | 83.2 | **84.2** |
| | 80% | 81.8 | 80.2 | 81.3 | 80.5 | 80.8 | 82.0 | 83.1 | 81.9 | 83.5 | **84.0** |
| | 90% | 82.7 | 81.9 | 80.8 | 80.3 | 81.7 | 82.3 | 83.2 | 80.9 | 83.2 | **84.4** |
| C-MHAD | 20% | 75.7 | 75.2 | 75.6 | 75.3 | 75.0 | 75.2 | 76.2 | 75.9 | 76.4 | **76.7** |
| | 40% | 74.4 | 75.0 | 74.7 | 74.2 | 74.0 | 74.1 | 74.9 | 74.6 | 75.1 | **75.8** |
| | 60% | 73.7 | 74.2 | 73.9 | 73.5 | 72.7 | 73.4 | 74.0 | 73.7 | 74.5 | **75.6** |
| | 70% | 74.2 | 74.7 | 74.4 | 74.0 | 70.4 | 73.9 | 74.2 | 72.2 | 73.9 | **76.1** |
| | 80% | 74.7 | 75.2 | 74.9 | 74.5 | 73.3 | 74.4 | 74.8 | 72.8 | 75.3 | **77.2** |
| | 90% | 74.1 | 74.6 | 74.0 | 73.9 | 73.1 | 73.8 | 74.5 | 73.1 | 75.4 | **77.3** |

**Table 2: Performance results for the two-modality missing scenario.**

| Modality Type | Method | 40% | 60% | 80% |
|---|---|---|---|---|
| Audio and Radar | FedMM+ZP | 69.8 | 58.5 | 60.2 |
| | FedMM+RP | 69.2 | 68.5 | 61.3 |
| | FedProx+ZP | 69.3 | 59.8 | 60.3 |
| | FedProx+RP | 68.9 | 70.7 | 60.9 |
| | FedBN+ZP | 68.7 | 59.1 | 59.3 |
| | FedBN+RP | 68.2 | 68.2 | 59.6 |
| | PmcmFL | 69.8 | 71.3 | 61.7 |
| | Harmony | 69.2 | 72.7 | 60.7 |
| | AutoFed | 69.4 | 68.5 | 61.2 |
| | **Ours** | **70.0** | **77.7** | **62.1** |
| Audio and Depth Image | FedMM+ZP | 82.3 | 79.3 | 79.2 |
| | FedMM+RP | 78.8 | 81.8 | 79.5 |
| | FedProx+ZP | 82.1 | 80.3 | 80.2 |
| | FedProx+RP | 81.4 | 82.0 | 79.8 |
| | FedBN+ZP | 81.8 | 80.8 | 78.7 |
| | FedBN+RP | 77.3 | 79.2 | 78.9 |
| | PmcmFL | 83.1 | 82.0 | 81.6 |
| | Harmony | 82.4 | 78.8 | 81.4 |
| | AutoFed | 82.8 | 79.4 | 81.0 |
| | **Ours** | **83.3** | **82.3** | **82.3** |
| Radar and Depth Image | FedMM+ZP | 33.1 | 37.9 | 28.5 |
| | FedMM+RP | 38.3 | 35.4 | 27.9 |
| | FedProx+ZP | 33.0 | 37.5 | 28.2 |
| | FedProx+RP | 36.2 | 36.9 | 27.8 |
| | FedBN+ZP | 32.7 | 37.2 | 27.9 |
| | FedBN+RP | 35.3 | 35.1 | 28.2 |
| | PmcmFL | 36.5 | 38.0 | 28.4 |
| | Harmony | 35.9 | 37.4 | 30.8 |
| | AutoFed | 37.6 | 38.7 | 31.1 |
| | **Ours** | **39.1** | **41.3** | **32.1** |

**Table 3: Cost comparison of different methods.**

| Method | FedMM | PmcmFL | FedProx | FedBN | Ours |
|---|---|---|---|---|---|
| GPU Usage (%) | 16 | 15 | 21 | 21 | 18 |
| Local Running Time (s) | 11.65 | 12.86 | 12.23 | 11.57 | 12.12 |

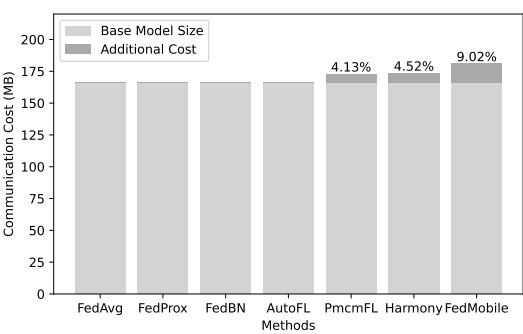

**Figure 3: Communication cost of FedMobile and baselines.**

both the modality missing rate and the number of missing modality types to control heterogeneity in multimodal FL. The primary reason for this adjustment is that multimodal FL involves multiple types of data, making it difficult for the Dirichlet function to reasonably partition the data. We further define two heterogeneous scenarios: (1) scenarios with different distributions of the number of missing modality types and (2) scenarios with varying distributions of modality missing rates, as illustrated in Figs. 7–8. In Scenario 1, we set the modality missing rate at $\beta = 40\%$ and randomly omit different numbers of modality types. The experimental results, summarized in Table 4, represent the average of five repeated trials. These results indicate that FedMobile consistently outperforms the other baselines, demonstrating its robustness in handling dynamic modality-missing scenarios. For Scenario 2, we control the number of missing modality types but dynamically adjust the modality missing rate, ranging from a maximum of $\beta = 80\%$ to a minimum of $\beta = 20\%$. The corresponding results, also presented in Table 4, show that FedMobile again outperforms the baselines, highlighting its strong performance even in these challenging non-IID settings.

**Large Node Scenario (RQ4).** To address RQ4, we investigate the performance of FedMobile and its baselines on large-scale nodes. While the dataset in RQ1 was collected from 10 nodes, we use the FLASH dataset, which involves 210 nodes, to more effectively partition the multimodal data and validate their performance.

uate the performance of the baselines and FedMobile in a dynamic modality-missing scenario on the ADM dataset. Unlike unimodal FL, where the Dirichlet function is commonly used to control the degree of heterogeneity (*i.e.*, non-IID data), we dynamically adjust

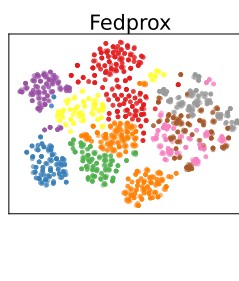 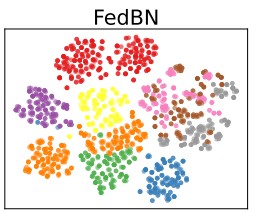 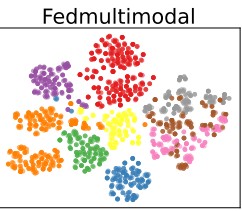 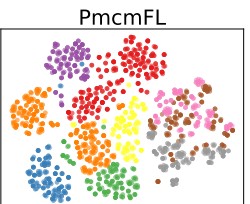 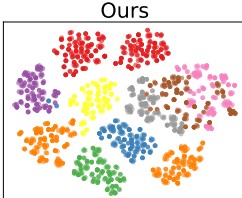

Fedprox  FedBN  Fedmultimodal  PmcmFL  Ours

Figure 4: Feature visualization results of different methods.

Table 4: Performance results in the heterogeneity scenario.

| Dataset | Method | Scenario 1 | Scenario 2 |
|---------|--------|------------|------------|
| **ADM** | FedMM+ZP | 69.3 | 71.4 |
| | FedMM+RP | 69.5 | 72.1 |
| | FedProx+ZP | 70.2 | 73.7 |
| | FedProc+RP | 69.5 | 73.2 |
| | FedBN+ZP | 69.8 | 71.3 |
| | FedBN+RP | 70.3 | 71.7 |
| | PmcmFL | 71.5 | 74.5 |
| | Harmony | 70.7 | 76.8 |
| | AutoFed | 70.1 | 77.9 |
| | **Ours** | **76.5** | **80.2** |

Table 5: Performance results on FLASH Dataset.

| Dataset | Method | 40% | 60% | 80% |
|---------|--------|-----|-----|-----|
| **FLASH** | FedMM+ZP | 52.7 | 51.1 | 50.4 |
| | FedMM+RP | 53.4 | 52.3 | 51.2 |
| | FedProx+ZP | 49.7 | 50.1 | 49.2 |
| | FedProx+RP | 49.4 | 48.7 | 49.4 |
| | FedBN+ZP | 49.9 | 49.1 | 47.8 |
| | FedBN+RP | 50.4 | 49.7 | 48.2 |
| | PmcmFL | 52.4 | 51.6 | 50.4 |
| | Harmony | 55.8 | 54.7 | 54.1 |
| | AutoFed | 54.9 | 53.4 | 55.4 |
| | **Ours** | **57.6** | **57.1** | **56.8** |

Specifically, we set modality missing rates at 40%, 60%, and 80%, and conduct five repeated experiments to record the average model accuracy. The results, presented in Table 5, show that FedMobile consistently outperforms the other advanced baselines even on large-scale nodes, demonstrating that its performance is not limited by the scale of the mobile node.

**Feature Visualization (RQ5).** In response to RQ5, we undertake a qualitative evaluation of the multimodal features generated by the competing methods by visualizing them. For this purpose, we employ the t-distributed Stochastic Neighbor Embedding (t-SNE) technique on dataset MHAD to project the high-dimensional multimodal features extracted by each method onto a lower-dimensional space. The resulting dimensionality reduction is presented in Fig. 4. Our visualization results indicate that FedMobile excels at extracting more precise and refined multimodal features, which in turn leads to enhanced classification accuracy. In comparison, alternative methods exhibit substantial deficiencies in feature extraction. This

Table 6: Numerical results of ablation experiments.

| Method | MHAD | USC | ADM | CMHAD | FLASH |
|--------|------|-----|-----|-------|-------|
| Ours (w/o $\mathcal{L}_{Train}$) | 74.7 | 57.5 | 79.8 | 79.2 | 52.7 |
| Ours (w/o SV) | 77.3 | 61.6 | 81.8 | 82.7 | 56.9 |
| Ours (w/o $\alpha$) | 74.5 | 58.6 | 84.1 | 77.4 | 53.8 |
| **Ours** | **77.9** | **62.8** | **84.3** | **75.8** | **57.6** |

observation underscores the value of FedMobile's dual strategies of local-to-global knowledge transfer and modal feature reconstruction, which collectively facilitate the effective exploitation and extraction of information from incomplete modal sources.

**Ablation Studies (RQ6).** To investigate RQ6, we conduct a systematic dissection of FedMobile by analyzing the performance contributions of its constituent parts with $\beta = 40\%$. To this end, we experimentally validate the performance of three ablated versions of FedMobile on four benchmark multi-modal datasets. Specifically, we successively deactivate the modal reconstruction network, the contribution sensing module, and the dynamic parameter aggregation module, forming three distinct variations of FedMobile. The experimental outcomes are summarized in Table 6, demonstrating that the modal reconstruction network and the contribution sensing module play pivotal roles in determining FedMobile's performance. On the other hand, the impact of the dynamic parameter aggregation module on FedMobile's performance appears to be less pronounced. For illustration, when the modal reconstruction network is removed from FedMobile, the performance degradation on the MHAD dataset reaches 3.2%, relative to the complete version of FedMobile. These findings highlight the critical importance of the modal reconstruction and contribution sensing mechanisms within the FedMobile framework.

## 6 Conclusion

The paper addresses the challenge of incomplete modalities in multimodal Federated Learning systems by proposing a new framework called FedMobile. Unlike existing methods that rely on unimodal subsystems or interpolation, FedMobile leverages cross-node multimodal feature information for reconstructing missing data and employs a knowledge contribution-aware mechanism to evaluate and prioritize node inputs, improving resilience to modality heterogeneity. The framework demonstrates superior performance in maintaining robust learning under significant modality loss compared to current standards, all while not increasing computational or communication costs. Overall, FedMobile represents a significant step forward in developing more efficient and resilient multimodal FL systems.

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

## A  Privacy Analysis

The proposed method for imputing missing modalities in FedMobile introduces privacy-preserving strategies by utilizing latent space reconstruction and conditional distributions across nodes. This privacy analysis explores how these strategies help mitigate privacy risks associated with multimodal FL under scenarios where data modalities are incomplete across nodes.

**Privacy Issues in Multimodal FL.**  On the one hand, in traditional FL, model updates can inadvertently leak sensitive information about the local datasets, especially when gradients are shared directly [49]. Furthermore, when reconstructing missing modalities from available data, there is a risk that sensitive or private information about the original data can be exposed [7]. High-dimensional data spaces are particularly vulnerable to this risk.

**FedMobile's Privacy-Preserving Mechanisms.**  FedMobile addresses these privacy risks through two key techniques:

• Conditional Distribution in Latent Space. Instead of directly imputing missing modalities in the raw input space $\mathcal{X}_k$, FedMobile reconstructs an induced distribution $\psi_k$ over a latent space $\mathcal{Z}_k$, as shown in Eq. (5). The latent space is more compact and lower-dimensional than the raw data space. This shift to a latent space significantly reduces the risk of privacy leakage because the latent representations contain abstracted information rather than raw, potentially sensitive features of the original data. Additionally, latent spaces typically obscure fine-grained details about individual data points, making it harder to reverse-engineer or infer private information from the shared model updates.

• Conditional Distributions for Imputation. FedMobile uses conditional distributions $Q_k$ (in Eq. (4)) and $\psi_k$ (in Eq. (5)) to model the relationships between the missing and present modalities. This distribution-based approach means that only the learned relationships between modalities are shared, not the actual data or detailed feature information. By focusing on conditional probabilities $p(y \mid x)$ or $p(y \mid z)$, the model implicitly encodes privacy since no raw features or labels are directly shared between nodes or with the central server. Instead, only probabilistic inferences are utilized, reducing the risk of reconstructing sensitive raw data.

Furthermore, existing work [33] has shown that it is difficult to recover training data by only obtaining gradient or feature information, as gradient leakage attacks are less effective on large training batches (*e.g.*, batch size = 32). Additionally, gradient leakage or feature reconstruction attacks are typically effective for image data [33, 49], but their effectiveness is limited for data types such as radar and gyroscope data.

## B  Proposed Algorithm

An overview of the algorithm is as follows:

## C  Additional Information

### C.1  Dataset Information

**MHAD Dataset.**  The MHAD dataset is designed to support research in human action recognition using multiple modalities. It includes data from 12 subjects performing 11 actions such as jumping, walking, running, and more. The dataset captures data from multiple sensors including accelerometers, gyroscopes, and

---

**Algorithm 1:** Description of the steps of the *FedMobile*.

**Input:** Local model $\omega_k$, local generator $\omega_G$, and local multimodal dataset $\mathcal{D}_k$, and validation dataset $\mathcal{D}_{val}$.

**Output:** Global model $\omega$

1  The server initializes the generator and global model and sends them to each node

   /* Knowledge Distillation-driven Cross-node Modalitiy Reconstruction Network     */

2  **while** *local training* **do**

3     Use the local model $\omega_k$ to perform feature extraction on the complete modality

4     Generate random label $y_r$ from $\mathcal{D}_{val}$

5     Input the random label $y_r$ and the missing modality label $y^m$ to the local generator $\omega_G$ to generate the corresponding features

6     Calculate $J(\omega_G^k)$ to optimize the local generator ▷Refer to Eq. (6)

7     Calculate $\mathcal{L}_{KL}^{m_0}(\omega_G^k; \omega_k)$ and $\mathcal{L}_{KL}^{m_1}(\omega_G^k; \omega_k)$ to further align the features of the missing modality ▷Refer to Eqs. (7)–(8)

8     Perform coupled training of local model and local generator via Eq. (9)

9  **end**

10  The nodes upload $\omega_k$ and $\omega_G^k$ to the server

   /* – Server does: –     */

   /* Clustered Shaple Value-driven Generator Contribution Evaluation Module     */

11  **do**

12     Use the K-means algorithm to cluster the features of random labels $y_r$ extracted by the local generator $\omega_G^k$

13     Perform average aggregation on local generator parameters $\omega_G^k$ in each cluster

14     Use SV to calculate the marginal contribution to the global model for the obtained representative nodes and use it as an aggregate weight ▷Refer to Eq. (11)

15     Aggregate these generators using the aggregation weights obtained from the above steps to produce high-quality generators ▷Refer to Eq. (12)

16  **while** *excute the above module*

   /* Contribution-aware Aggregation Rule     */

17  **do**

18     Calculate the local and global contributions of the node via Eq. (13) and Eq. (15)

19     Calculate the adaptive weight $\alpha_k$ via Eq. (18)

20     Perform contribution-aware aggregation via Eq. (19) to obtain global model $\omega$

21  **while** *excute the above module*

22  **return** The optimal global model $\omega$.

---

magnetometers, as well as from optical motion capture systems and video cameras. Link: https://paperswithcode.com/dataset/berkeley-mhad

Table 7: Summary of the four multimodal datasets.

| Dataset | Modality Information | # Classes | # Users | Object | Domain | # Samples | Size (MB) |
|---|---|---|---|---|---|---|---|
| USC-HAD | Acc, Gyro | 12 | 10 | People | Activity Detection | 38312 | 38.5 |
| MHAD | Acc, Skeleton | 11 | 10 | People | Activity Detection | 3956 | 187 |
| ADM | Audio, Radar, Depth Image | 11 | 10 | People | Medical | 22452 | 30208 |
| C-MHAD | Video, Inertial Sensor | 7 | 10 | People | Activity Detection | 7802 | 24268.8 |
| FLASH | GPS, LiDar, Camera | 64 | 210 | Traffic Scenes | Autopilot | 32923 | 5232.64 |

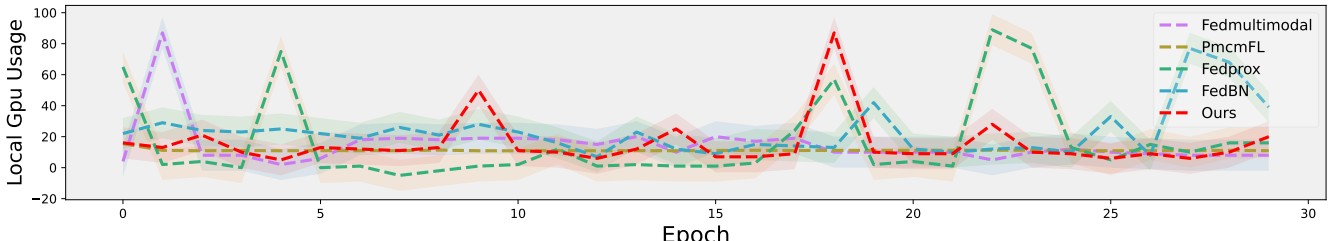

Figure 5: Numerical result of local GPU usage.

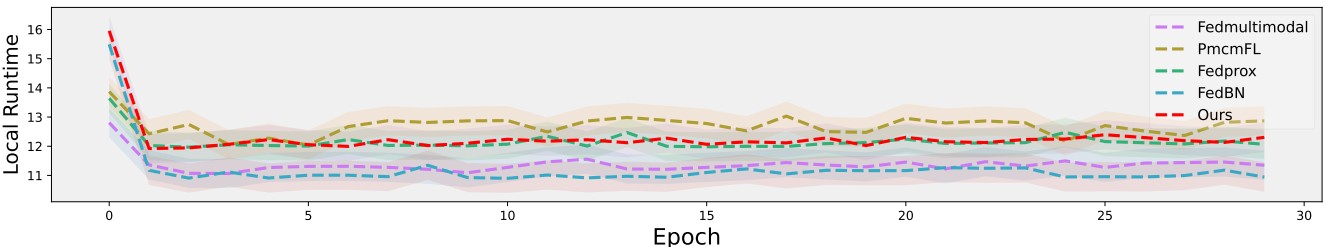

Figure 6: Numerical result of local running time.

**USC-HAD Dataset.** The USC-HAD dataset is a collection of data gathered for the purpose of recognizing human activities. The dataset includes data from 14 subjects performing 12 different activities such as walking, running, jumping, sitting, standing, and more. The data is captured using wearable sensors that record accelerometer and gyroscope readings. Link: https://sipi.usc.edu/had/

**ADM Dataset.** The ADM dataset focuses on detecting Alzheimer's disease by analyzing 11 behavioral biomarkers in natural home environments. These biomarkers include activities such as cleaning living areas, taking medications, using mobile phones, writing, sitting, standing, getting in and out of chairs/beds, walking, sleeping, eating, and drinking. The three modal data of depth images, radar, and audio are obtained by sampling from the depth camera, mmWave radar, and microphone at sampling rates of 15 Hz, 20 Hz, 44 Hz, and 100 Hz, respectively. Link: https://github.com/xmouyang/Harmony/blob/main/dataset.md

**C-MHAD Dataset.** The C-MHAD dataset extends the concept of the MHAD dataset by providing continuous recordings of human activities. Unlike datasets that capture discrete instances of actions, C-MHAD includes long, continuous streams of activity data, simulating real-world scenarios where actions flow into

one another without clear boundaries. This dataset is particularly useful for developing and testing algorithms that need to operate in real-time and handle continuous input, such as those used in surveillance, human-computer interaction, and assistive technologies. Link: https://github.com/HaoranWeiUTD/C-MHAD-PytorchSolution

**FLASH Dataset.** The FLASH dataset is a multimodal dataset designed specifically for multimodal FL in traffic scenarios. It includes 32,923 samples from three modalities, collected in real time from autonomous vehicles equipped with various sensors—GPS, LiDAR, cameras—and roof-mounted Talon AD7200 60GHz millimeter-wave radios. The dataset primarily supports research in autonomous driving and high-band millimeter-wave sector prediction, among other related fields. https://repository.library.northeastern.edu/files/neu:k930bx06g

## C.2 Data Partitioning

**IID Setting.** For the IID data setting, we assign the multimodal dataset collected by each mobile sensor to each node. In addition,

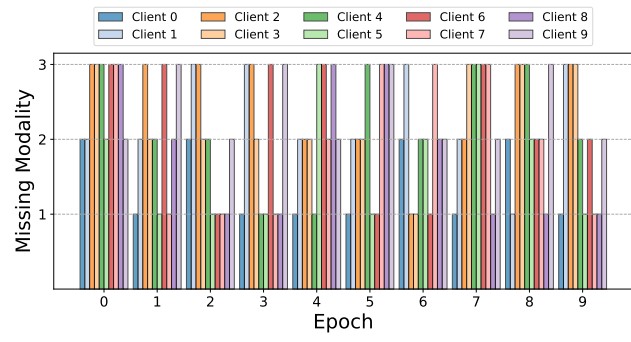

**Figure 7: Distribution of missing modality at different nodes.**

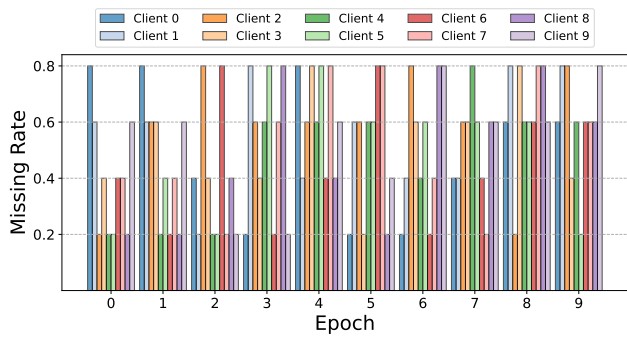

**Figure 8: Distribution of missing rate at different nodes.**

during training, we keep the type and missing ratio of each node's missing modality consistent to construct an IID data scenario.

**Non-IID Setting.** We define two non-IID data scenarios: (1) scenarios with different distributions of the number of missing modality types and (2) scenarios with varying distributions of modality missing rates, as illustrated in Figs. 7–8. In Scenario 1, we set the modality missing rate at 40% and randomly omit different numbers of modality types. For Scenario 2, we control the number of missing modality types but dynamically adjust the modality missing rate, ranging from a maximum of 80% to a minimum of 20%.

## C.3 Generator Information

The generator used in this paper is a multi-layer perceptron (MLP), which performs updates and optimizations in conjunction with local training (see Eqs. (4)-(9)). Specifically, the generator architecture comprises two fully connected layers, a batch normalization (BN) layer, and an activation layer. Initially, the first fully connected layer maps the data labels into feature embeddings. This is followed by the BN layer and a non-linear activation layer. Finally, the second fully connected layer serves as the representation layer, converting the feature embeddings into a format suitable for model training. While more complex generators can be used, the MLP is a good choice to minimize overhead.

