# OpenReview forum: "FedMobile: Enabling Knowledge Contribution-aware Multi-modal Federated Learning with Incomplete Modalities"
_ACM.org/TheWebConf/2025/Conference — WWW 2025 Poster_

### Official Review · Reviewer_2aXN · 2024-11-11

**Novelty:** 4
**Technical Quality:** 5

**Review:**

The authors present FedMobile, an effective Federated Learning approach for multimodal learning that can deal with missing modalities. Unlike existing approaches, the key idea of FedMobile is to train a feature generator that can generate missing modality data. The system consists of several other components: (1) a knowledge-distillation-based approach to impute missing modalities, (2) a contribution evaluation module to cluster generator contributions, using the Shapley Value, and (3) a node-level contribution-aware aggregation rule. The authors implement the system and present experimental results using different datasets and baselines.

The paper touches upon a timely and important topic as multimodal model training is an emerging area of research. Missing modalities is an adequate concern and I appreciate the authors’ efforts to design and evaluate an end-to-end system that addresses some of the issues arising while doing this.

The paper seems to be quite complete, containing an extensive set of evaluations using five datasets and six different baselines, with varying modality characteristics and scale in terms of network size. The authors evaluate different aspects of the approach, including overhead, performance and an ablation study. In all evaluated settings, FedMobile seems to outperform existing baselines. Furthermore, the paper is generally well-written.

That brings me to some points for improvement. The paper is dense, making it at times difficult to understand all equations and their interdependencies. I got a bit confused about some of the steps, and various steps have not been explained in much detail, for example, the modality alignment. While they have been explained through equations, an intuitive explanation is often missing. Some more high-level description of these aspects would have been helpful, although I understand that the rather short page limit imposed by the conference might make this difficult. I also struggle to see a clear connection between the proposed components, making the system appear somewhat as a disjoint (yet effective!) set of optimisations). So, while the idea appears to be novel and effective, the presentation of the algorithm could be improved.

A concrete suggestion to improve presentation would be to add step numbers to Figure 2 and explain each of these steps. In the same figure, you can also reference the section numbers in which the step is explained. It would also be helpful to emphasize in the textual description which steps are done by the server and which steps are executed by clients locally.

It was unclear to me what the authors exactly mean with a “missing modality”. This can be either on a node level or sample level. For instance, it could be that a particular node is missing data for a particular modality for all samples in its entire dataset (if a sensor has broken down entirely), or that only some samples of a particular node are missing a modality (if a sensor sometimes malfunctions). Could the authors clarify on this?

I would have appreciated some more details/insights on how the baselines are working and why they perform worse than FedMobile. The experiments do not provide many of such details, although this is most likely a consequence of the limited number of pages.

Minor comments:
- For completeness, consider adding the validation dataset to Figure 2.
- Please consider adding a label with beta to Table 3 and Table 5.
- Typo in the header of Section 4.2: “Shaple” -> “Shapley”.
- Adding a table with notation in the Appendix would have been helpful.

**Questions:**

1) What is the intuition behind using random labels y_r alongside the missing modality labels during the “knowledge distillation-driven cross-node modality reconstruction”?
2) Can the authors comment on how the knowledge distillation used in FedMobile is similar to or different from traditional knowledge distillation?
3) Are there any other limitations besides additional compute/computation costs of the proposed approach?
4) In Table 1, the accuracy of ADM seems to be quite stable under variance of beta compared to other datasets. Any insights on why this is the case?
5) I did not understand why AutoFed and Harmony (which seem to be the most relevant baselines) were excluded when answering RQ2 (Table 3 in particular). The authors mention that this is because they also include hardware equipment, but I’m not sure what is meant by this.
6) The authors consider a scenario where a modality is either there or missing. Would FedMobile still work if, say, a particular sensor produces wrong/invalid/out-of-distribution data?
7) Line 681 mentions that FedProx does not introduce much additional overhead, but as far as I’m aware, FedProx adds proximal term to the local loss function and therefore does not induce additional communication overhead?
8) The authors are using a proxy/validation dataset to generate random labels (see line 1105). Is this a shared validation dataset (between clients and between client-server), and how is this dataset constructed and distributed between nodes? This reliance on such a dataset in general might warrant some more discussion and motivation - do other baselines use such a dataset as well?
9) RQ2 aims to quantify the computational and communication overhead of FedMobile and baselines. Do the authors also have some numbers on the computational complexity of each other involved steps in FedMobile?

**Reviewer Confidence:**

3: The reviewer is confident but not certain that the evaluation is correct

**Scope:**

3: The work is somewhat relevant to the Web and to the track, and is of narrow interest to a sub-community

---

### Official Review · Reviewer_u9gC · 2024-11-25

**Novelty:** 5
**Technical Quality:** 5

**Review:**

This paper addresses the challenge of incomplete modalities in Multi-modal Federated Learning by leveraging model distillation and clustered Shapley value methods, achieving state-of-the-art performance.

**Strengths**

1. The paper is easy to follow.

2. It uses model distillation to address the issue of incomplete modalities in Multi-modal FL.

3. It is the first to employ knowledge contribution-aware aggregation and Shapley value to enhance performance.

**Weaknesses**

1. Placement of Tables and Figures: Tables 2–6 and Figures 3–4 should appear before their first reference in the main text. Additionally, Figures 7–8 are located in the appendix, and this should be explicitly stated in the main text.

2. FedAvg Experiments: The paper does not provide experiments on FedAvg, yet Figure 3 includes a comparison with FedAvg. The absence of such experiments is inconsistent.

3. Flash Dataset: Table 1 does not include experiments on the Flash dataset.

4. Model Details in Section 5.1: Section 5.1 does not clearly explain the downgraded models. For instance, the paper references 2D-CNN and 3D-CNN without sufficient detail. Also, the paper should conduct experiments with big models.

5. Local Running Time in Table 3: The use of the term “local running time” in Table 3 is misleading. This appears to refer to the aggregation time for the global model.

Since the method employs model distillation, the paper should also report the time and GPU utilization for local client-side operations.

6. Marginal Improvements: The performance improvement is not particularly significant. Model distillation is resource-intensive, and many existing methods, such as FedProx, effectively handle non-IID data with much simpler adjustments (e.g., using ZP or RP). The marginal improvement here raises concerns about the method’s practical efficiency.

7. Code Issues: Upon reviewing the provided code, I found several issues:
The code does not include an implementation of Shapley value (SV) and only provides basic modules. It lacks runnable scripts and proper installation dependencies. The code appears to support only three datasets, not the five datasets claimed in the paper. The authors should clarify and address these discrepancies.

**Questions:**

Refer to the weaknesses above.

**Reviewer Confidence:**

4: The reviewer is certain that the evaluation is correct and very familiar with the relevant literature

**Scope:**

4: The work is relevant to the Web and to the track, and is of broad interest to the community

---

### Official Review · Reviewer_VYQU · 2024-11-27

**Novelty:** 4
**Technical Quality:** 4

**Review:**

Existing multi-modal federated learning (FL) techniques often fail to account for the shared latent feature space between incomplete modalities across nodes and do not effectively distinguish low-quality nodes. To address these issues, the authors propose FedMobile, which prioritizes local-to-global knowledge transfer and utilizes cross-node multi-modal feature information to reconstruct missing data.

The framework appears particularly effective for web-based mobile sensing systems in the Web of Things (WoT), with key contributions including a knowledge distillation-driven cross-node modality reconstruction network and an efficient generator contribution evaluation module based on clustered shapley values and a contribution-aware aggregation mechanism.

In the introduction, it is necessary to include a clearer presentation of the challenges and specific solutions offered by FedMobile. It feels more natural to first outline the problem background, then the challenges and solutions, before summarizing contributions concisely. The current structure, which includes experimental details and baseline discussions in the contribution section, feels somewhat misplaced.

The distinction between unimodal and multimodal FL in Fig. 1 is difficult to distinguish. The figure is too small, and the visual representation of differences is not effective. Enhancing the figure with clearer annotations or additional explanations would help readers understand the contrasts more effectively.

The novelty of FedMobile’s contribution mechanism is somewhat unclear. Strategies for evaluating and adapting node contributions have been widely used in areas like transfer learning, personalized federated learning, and model generalization. Additionally, the paper refers to the 210-node FLASH dataset as "large-scale." However, in federated learning, much larger networks (e.g., 400+ nodes) are commonly used without being labeled as such. For example, in J. Mills et al. (2022), 400 clients are utilized without being described as "large-scale." Clarifying why 210 nodes are considered large-scale in this context would strengthen the claim.
In Section 5.2, the comparison with other methods, particularly FedMM, does not highlight clear distinctions. Including quantitative metrics or indicators for feature extraction would make the differences more apparent. Data heterogeneity, which is critical to federated learning experiments, is only briefly discussed in the appendix. Incorporating details about data partitioning in the main text, particularly within the experimental settings or RQ3, would improve the clarity and logical flow of the paper.

The improvement claims in the experimental results require further elaboration. The rationale behind the large-node scenario (RQ4) and its significance in this work should be better explained. Additionally, providing numerical comparisons and detailed discussions on performance improvements would help readers understand the practical implications of FedMobile.

**Questions:**

1. What makes FedMobile unique compared to existing approaches?
2. How can the performance improvements over the baselines be demonstrated as significant?
3. How does the proposed method address the challenges of data heterogeneity effectively?

**Reviewer Confidence:**

2: The reviewer is willing to defend the evaluation, but it is likely that the reviewer did not understand parts of the paper

**Scope:**

3: The work is somewhat relevant to the Web and to the track, and is of narrow interest to a sub-community

---

### Official Review · Reviewer_WwBc · 2024-11-30

**Novelty:** 5
**Technical Quality:** 5

**Review:**

1. Quality
The paper presents a robust framework for handling multimodal federated learning (FL) with incomplete modalities. The methodology combines innovative elements like knowledge distillation and Shapley value-based contribution evaluation. While the experimental evaluation is comprehensive, some theoretical components need more clarity, and integrating key concepts is not seamless. For example, Equation 6 is challenging to interpret due to insufficient explanation of terms, making it less accessible for readers.


2. Clarity

The paper is generally well-organized, with a clear flow from problem identification to methodology and results. However, the disconnection between the contribution-aware mechanism and the missing modality problem creates a lack of cohesiveness.
Additionally:
• The alignment of modalities does not consider cross-modality correlations, which could been utilized to improve the reconstruction accuracy by leveraging dependencies between present and missing modalities.
• The shared feature space, a key motivation for the method, is not validated, undermining the paper's claims.

3. Originality

The originality is commendable, as the framework introduces:
1. A cross-node modality reconstruction network leveraging shared feature spaces.
2. A contribution-aware aggregation mechanism using Shapley values.

However, the lack of strong validation for utilizing the shared feature space and the omission of cross-modality correlation modeling are notable limitations. While these concepts are novel, they require better integration and empirical substantiation.


4. Significance

The paper’s significance lies in addressing real-world challenges in privacy-sensitive and dynamic FL systems, such as mobile healthcare and autonomous driving. It provides a promising solution for incomplete modality scenarios, outperforming state-of-the-art methods. Nevertheless:
• The disconnection between components weakens the overall focus.
• The practical relevance of the contribution-aware mechanism in addressing missing modalities is not clearly justified.

5.
Pros
•	Robust Experimental Results: Comprehensive evaluations across five benchmark datasets demonstrate the framework’s effectiveness, even with up to 90% missing modalities. It consistently outperforms state-of-the-art methods.
•	Addressing Real-World Challenges: The method tackles practical issues like modality incompleteness and heterogeneous data distributions, which are common in mobile and IoT environments.
•	Scalable and Practical Design: By integrating contribution-aware aggregation and efficient reconstruction, the framework manages computational and communication overhead effectively, making it suitable for resource-constrained scenarios.
•	Significant Performance Gains: The proposed method achieves higher accuracy and robustness compared to existing approaches, especially in scenarios with dynamic or severe modality loss.

Cons
•	The shared feature space is emphasized due to it was overlooked by existing literature. Still, there’s no direct evidence to prove that, and also to show how effectively it’s being utilized in this paper. The paper lacks experiments or visualizations to validate this claim, leaving the argument incomplete.
•	The contribution-aware mechanism doesn’t seem directly related to solving the missing modality problem. This makes the framework feel disjointed, as if it’s tackling two loosely connected problems rather than one cohesive challenge.
•	Unclear Presentation of Key Equations: Equation 6 is hard to follow due to insufficient explanations of the terms and their significance. Readers are left guessing about its implementation, which could hinder reproducibility.
•	No Consideration of Cross-Modality Correlations: The alignment process doesn’t factor in how present modalities might relate to missing ones. Ignoring these correlations could lead to suboptimal feature reconstruction, especially when strong dependencies exist between modalities.

**Questions:**

Please provide more evidence or experiments to back up the claim about the shared feature space. As you mentioned, existing methods overlook this.

How exactly does the contribution-aware mechanism help address missing modalities? Is there any data showing that prioritizing high-quality nodes directly improves the accuracy of the reconstructed features?

Please explain the terms in Equations 6-8 more clearly. And give more detail on how the feature generator is designed.

How does this method perform in networks with thousands of nodes? Currently, the nodes are only set to 10, which is far from practice. Are there specific challenges or bottlenecks you’d expect to face in those larger setups?

**Reviewer Confidence:**

4: The reviewer is certain that the evaluation is correct and very familiar with the relevant literature

**Scope:**

3: The work is somewhat relevant to the Web and to the track, and is of narrow interest to a sub-community